# Automated detection of COVID-19 through convolutional neural network using chest x-ray images

**Rubina Sarki**[1]*, **Khandakar Ahmed**[1], **Hua Wang**[1], **Yanchun Zhang**[1], **Kate Wang**[2]

**1** Institute for Sustainable Industries & Liveable Cities, Victoria University, Melbourne, Victoria, Australia,
**2** RMIT, Melbourne, Victoria, Australia

* rubina.sarki@live.vu.edu.au

## Abstract

The COVID-19 epidemic has a catastrophic impact on global well-being and public health. More than 27 million confirmed cases have been reported worldwide until now. Due to the growing number of confirmed cases, and challenges to the variations of the COVID-19, timely and accurate classification of healthy and infected patients is essential to control and treat COVID-19. We aim to develop a deep learning-based system for the persuasive classification and reliable detection of COVID-19 using chest radiography. Firstly, we evaluate the performance of various state-of-the-art convolutional neural networks (CNNs) proposed over recent years for medical image classification. Secondly, we develop and train CNN from scratch. In both cases, we use a public X-Ray dataset for training and validation purposes. For transfer learning, we obtain 100% accuracy for binary classification (i.e., Normal/ COVID-19) and 87.50% accuracy for tertiary classification (Normal/COVID-19/Pneumonia). With the CNN trained from scratch, we achieve 93.75% accuracy for tertiary classification. In the case of transfer learning, the classification accuracy drops with the increased number of classes. The results are demonstrated by comprehensive receiver operating characteristics (ROC) and confusion metric analysis with 10-fold cross-validation.

**Data Availability Statement:** 1) COVID-19 Image Data Collection: Prospective Predictions Are the Future, Joseph Paul Cohen and Paul Morrison and Lan Dao and Karsten Roth and Tim Q Duong and Marzyeh Ghassemi arXiv:2006.11988, https://

## 1 Introduction

The novel coronavirus (COVID-19) is an infectious disease which started late December 2019 and has spread across the world. The World Health Organization (WHO) announced a COVID-19 as a pandemic on the 11th of March 2020. This epidemic continues to have a catastrophic impact on health and wellbeing worldwide. A critical step in the COVID-19 combat cycle is to develop an efficient classification system so that patients can begin to receive prompt medical care, treatment, and control transmission.

During this short time, many researchers have attempted to develop various screening tools and classification systems. For example, reverse transcriptase-polymerase chain reaction (RT-PCR) is the critical screening tool to detect severe acute respiratory syndrome (SARS)-COV-2 [1] and as well as COVID-19. While the RT-PCR test is the standard screening tool to

github.com/ieee8023/covid-chestxray-dataset, 2020 2) https://www.kaggle.com/paultimothymooney/chest-xray-pneumonia, and https://www.cell.com/cell/fulltext/S0092-8674(18)30154-5.

**Funding:** The authors received no specific funding for this work.

**Competing interests:** The authors have declared that no competing interests exist.

detect COVID-19, it also has limitations. The procedure of RT-PCR is very complicated and also time-consuming [2–4]. Therefore, attempts have been made to diagnose COVID-19 through chest radiography imaging such as computed tomography (CT) or chest x-ray images. Tao et al. [2], for example, reported the diagnostic importance and accuracy of CT chest images over RT-PCR in COVID-19. Their findings show that a chest CT has a high sensitivity for the diagnosis of COVID-19.

On the other hand, Guan et al. [5], reported radiographic abnormalities of positive COVID-19 cases such as interstitial abnormalities, bilateral abnormalities, and ground-glass opacity in both CT and chest x-ray images. Although most of the previous discussion has focused on CT scan imaging, resulting in increased image specificity in acquisitions, there are many benefits using chest x-ray imaging for COVID-19 monitoring. The downside of CT imagery is processing time. Also, good high-quality CT scanners are not available in many developing countries; in that case, timely scanning of COVID-19 is impossible. On the other hand, chest x-rays are the most viable and are generally available clinical imagery method, playing a significant role in primary care and observational studies [6, 7].

CNN has been explored extensively in the field of COVID-19 classification and detection [8, 9], largely exceeding previous techniques for image recognition [10]. Overall, CNN has illustrated enormous healthcare capacity to classify patients at higher risk of developing a disease. The application of CNN ranges from binary classification to multi-class classification. CNNs have already shown good results in discovering the intricate structures in high-dimensional datasets, with multi-layer function representations.

However, timely detection of COVID-19 with high classification accuracy and minimal data is still an open challenge. The quantity of annotated data for training and data quality are two key factors while building a detection system. Chest x-ray images obtained from publicly available dataset for experiments shown in Fig 1 are limited in numbers. Hence, due to the limited volume of COVID-19 data samples, transfer learning is considered a suitable approach for classification purposes. In transfer learning, transfer of learned parameters from a source task to a target task is the key to achieving the highest accuracy in a limited dataset. Transfer learning results are encouraging and have illustrated the effectiveness of deep learning (DL) networks in binary classification [11]. To evaluate the model's effectiveness in the diagnosis of COVID-19, we performed various experiments. The pipeline process is illustrated in Fig 2 and the primary contributions of this research are summarized as follows:

- We propose CNNs (VGG16, InceptionV3, Xception) transfer learning based models in two classification scenarios; (i) Scenario I consists of Normal/COVID-19 classification and (ii) Scenario II consists of Normal/COVID-19/Pneumonia classification.

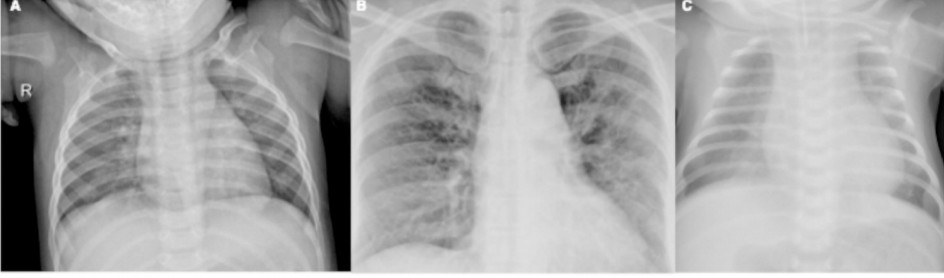

**Fig 1. Chest x-ray images: (A) normal; (B) COVID-19 positive; (C) viral pneumonia.**

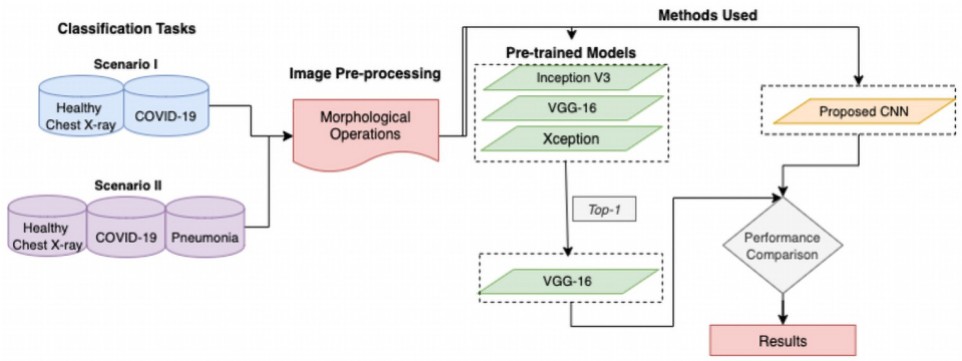

**Fig 2. The pipeline process.**

- We have proposed an innovative five-layers CNN architecture to classify the scenarios to improve performance accuracy.

- Our experiment results demonstrate the effectiveness of models in classification scenarios and their potential for COVID-19 classification, detection, prevention, and control.

The organization of this paper is as follows. Section I presents the literature review with surveying articles. Section II includes all required materials and associated methods for the research in this paper. Section III addresses the results obtained and comparisons. Section IV discusses the developed methods, performances and automated COVID-19 Detection. Finally, section V concludes this article and future work.

## 2 Background

The demand for faster diagnosis of COVID-19, multiple studies implemented to highlight design solutions and scientific facts regarding this highly infectious disease. Some image identification, analysis, interpretation, and decision-making techniques are listed in this section. Deep Learning (DL) [12] has been proposed and has successfully acquired promising results in terms of accuracy in various fields [11, 13–17]. Case studies on COVID-19 screening of CT images were presented by authors such as Xu et al [18], Gozes et al. [19], Li et al. [20], Shi et al. [21]. Authors Xu et al. [18] discussed that COVID-19 exhibits its characteristics that vary from other forms of viral pneumonia, like viral influenza-A pneumonia. The study's objective was to develop an initial screening framework for COVID-19 by automated pulmonary CT images of COVID-19, pneumonia, and normal cases. They employed 618 CT scan sample images before augmentation, and their model obtained an accuracy of 89.7%. The author's method includes image pre-processing, segmentation of the multiple regions (patches) adopting volumetric network (V-Net) [22] bases segmentation model V-Net-IR-RPN [23], which trained for pulmonary tuberculosis purpose. Finally, classification was performed by ResNet using segmented patches of COVID-19, pneumonia, and Normal images. Shan et al. [24] proposed a method for the automated segmentation and quantification of infection in COVID-19 patient CT scans. The data collected consisted of 549 CT images. A Human-In-The-Loop (HITL) technique was introduced to facilitate the manual delineation of CT images for processing. This allows the generation and enrichment of a training set provided as input to an ML system operating over the infected COVID-19 region to an automatic segmentation stage. The mechanism repeats these measures from these auto-contoured areas to support radiologists in their

refining of annotations. The proposed model yielded classification accuracy of approximately 91.6% between automated and manual segmentation approach, and an average mean percentage of infection (POI) error of 0.3% for the entire lung on the validation dataset. Manual classification often takes 1-5 hours; the HITL approach significantly decreases the categorization time to four minutes after three model upgrade iterations.

Ng et al. [25] stated that pulmonary infections could be more clearly visible in CT images than in x-ray images of the chest. However, COVID-19 using chest x-ray images implemented as they represent comprehensive resources that are often analyzed upstream of CT scans. Recently, an initiative by Cohen et al. [26] to provide a repository comprising of COVID-19 positive, Middle East respiratory syndrome (MERS), Acute respiratory distress syndrome (ARDS), and SARS cases with annotated CT and chest x-ray images, so that the research group and community data scientists can use the dataset to analyze and develop AI systems for COVID-19 detection. After the immediate public disclosure of the suggested COVID-19 data, several automated diagnostic systems have been designed using a chest x-ray image [27–35]. Most of the studies have adopted DL-based architecture for developing the COVID-19 diagnostic tool and achieved promising accuracy. Thus, this is significant to note that the COVID-19 dataset continues to grow as new patient cases are continually growing and making publicly accessible regularly. Wang et al. [9] introduced a DL architecture called COVID-19 in which authors utilize the open dataset of the Chest x-ray images (Pneumonia) and the COVID-19 public dataset by [26]. The author's derived chest x-ray dataset, called COVIDx, comprises of 5941 posteroanterior chest radiography in 2839 patient cases. Their analysis targets four image categories: healthy, bacterial-infection, non-COVID viral-infection, and COVID-19 viral-infection. The dataset includes 1203 patients as healthy, 931 patients with bacterial pneumonia, 660 patients with non-COVID-19 viral pneumonia, and 68 x-rays from 45 patients with COVID-19. The authors use the principles of residual architecture design by He et al. [36], they utilize generative synthesis by Wong et al. [37] a machine-driven strategy for developing the final COVID-Net network topology that achieves 83.5% as global test accuracy. However, this result was performed, including small sample data relating to only ten COVID-19 cases.

## 3 Materials and methods

The research's overarching objective is to evaluate the performance of automated detection of COVID-19 from Chest x-ray images through an empiric assessment of classification improvement techniques. An experiment was performed using two sources of x-ray dataset. The associated objectives for this research identified as follows:

- Building a robust framework for COVID-19 classification using a chest x-ray.

- Classify chest x-ray images with proposed CNN and 3 pre-trained models.

- Evaluate and Compare the performance of the models with performance metrics.

### 3.1 Data collection

Initially, the data repository by Cohen et al. [38] was analyzed to collect COVID-19 images. A set of x-ray images collected from Cohen et al. [26]. The dataset contains 296 images in which 83 female and 175 male positive COVID-19 cases. Not all patient's details are provided with complete metadata in this data set. Further, bacterial pneumonia x-ray images by Kermany et al. [39] were obtained for COVID-19 and Pneumonia's classification. In particular, we merged and updated the two data repositories to create the experiment dataset. The gathered data consists of 1341 healthy images, 296 images with positive and suspected COVID-19, and

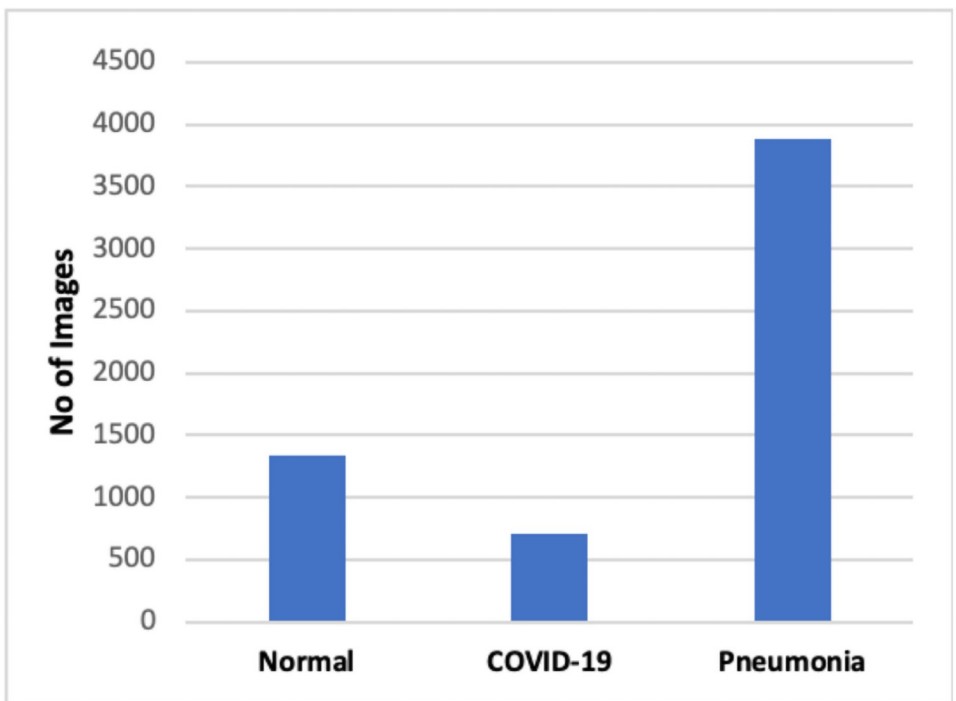

**Fig 3. Variation in chest x-ray images distribution.**

3875 images with viral and bacterial Pneumonia positive images. Therefore, data imbalance can be observed in the gathered data shown in Fig 3, which can give misleading classification results. Therefore, we examined all the images manually and remove overexposed, underexposed images. Finally, we selected 140 images from each category for our experiments.

## 3.2 Convolutional neural network

Recently, CNNs are the most studied machine learning (ML) algorithms for medical lesions diagnosis using images [40]. The justification behind this is that CNNs retain complex features when scanning input images. As stated above, spatial relationships are of primary importance in radiology, such as how the bone joins the muscle, or where standard lung tissue interfaces with infected cells. The system architecture is illustrated in Fig 4 and selected hyperparameters are shown in Table 1. This proposed CNN architecture has five convolution layers that take a chest image tensor of $244 \times 244$ as its input. Subsequently, the first convolution layer uses $5 \times 5 \times 3$ kernel filters with stride $1 \times 1$, and a total of 64 such filters are employed. The next layer,

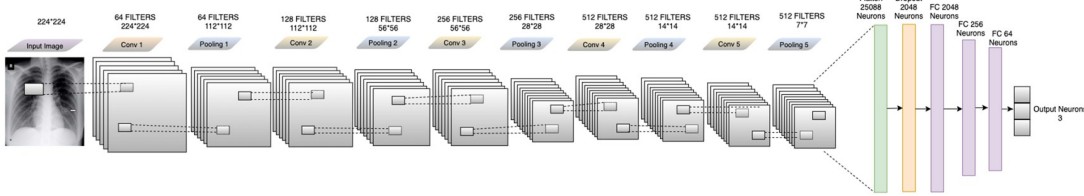

**Fig 4. Feature extraction of the input image is performed via the convolution, ReLU and pooling layers, before classification by the fully connected layer.**

**Table 1. Hyper-parameters of the build CNN model and preferred weights in this study.**

| R1 | R2 | R3 | R4 | R5 | R6 | R7 | R8 |
|---|---|---|---|---|---|---|---|
| **CNN** | 224*224 | RMSprop | 32 | 10-fold | 3e-4 | BCE CCE | 50 |

R1—Model, R2—Image Size, R3—Optimizers, R4—Mini Batch Size, R5—cross validation, R6—Initial Learning Rate, R7—Loss function, R8—Epoch, BCE—Binary cross-entropy, CCE—Categorical cross-entropy.

which receives the output from the first layer, is a max-pooling layer with $2 \times 2$ stride, reducing the input to half of its size $112 \times 112$. For all layers, the output from the pooling layer passes through the ReLU activation feature. The nonlinear output obtained now fed into the next convolution layer with $5 \times 5 \times 64$ with 128 filters, and the stride value is the same $1 \times 1$. The obtained output pass through a max-pooling layer with the same $2 \times 2$ strides, which again reduced the input to half of its size $56 \times 56$. After the output pass through ReLU activation, it is fed into the third convolution layer with 256 filters and the kernel size $5 \times 5 \times 128$ with $1 \times 1$ stride. The output is passed to a max-pooling layer, which results in a tensor of shape $28 \times 28$. Again the output pass through ReLU activation, fed into the fourth convolution layer with 512 filters and kernel size $5 \times 5 \times 256$ and with the same stride $1 \times 1$. The output from the fourth convolution is max-pooled to a size of $14 \times 14$. After ReLU activated and it is pass to a fifth convolution layer with 512 filters and $14 \times 14 \times 512$ kernel size to accommodate the output of all the filters from previously configured layers, and max-pooling of output from that layer with a stride of size $2 \times 2$ produces an output of size $14 \times 14$. Now the resulting tensor has the shape $7 \times 7 \times 512$. The obtained tensor is flattened with 25,088 neurons. The weighed values that emerge as neurons demonstrate the proximity to the symptoms of COVID-19. The drop-out layer is applied here to drop values to handle network overfitting. In our work, we used a dropout rate of 0.5 during training. The fully connected layer converts the tensor with 25,088 neurons to 64 neurons and adds ReLU activation to the output. A tensor with 64 neurons is the product of the fully connected layers; these 64 neurons are translated into neuron counts equal to the number of categories to which the retinal image belongs, healthy, COVID-19, and pneumonia.

**3.2.1 Convolution layer.** This layer comprises a filter set (kernel). Each filter is convoluted against the input image and then extract features by creating a new layer. Each layer signifies some of the important features or characteristics of the input image. The $^{*}$ symbol identifies the operation of the convolution. The output (or function map) $F(t)$ is defined below when input $I_n(t)$ is convoluted with a filter or $f(a)$ kernel.

$$F(t) = (I_n * f)(t). \tag{1}$$

If $t$ can only accept integer values, the following discrete convolution is provided by the following equation:

$$F(t) = \sum_a I_n(a) \cdot f(t - a). \tag{2}$$

The above assumes a one-dimensional convolutional operation. A two dimension convolution operation with input $I_n(m, n)$ and a kernel $f(a, b)$ is defined as:

$$F(t) = \sum_a \sum_b I_n(a, b) \cdot f(m - a, n - b). \tag{3}$$

By the commutative law, the kernel is flipped and the above is equivalent to:

$$F(t) = \sum_a \sum_b I_n(m - a, n - b) \cdot f(a, b).$$

(4)

Neural networks implement the cross-correlation function, which is the same as convolution but without flipping the kernel.

$$F(t) = \sum_a \sum_b I_n(m + a, n + b) \cdot f(a, b).$$

(5)

**3.2.2 Rectified Linear Unit (ReLU) layer.**   This layer is an activation function that sets the negative input value to zero, which optimizes and speeds up analyses and training, and helps prevent the gradient from disappearing. Mathematically, this described as:

$$R(x) = max(0, x).$$

(6)

In which x is input to the neuron.

**3.2.3 Maxpooling layer.**   This Layer is a sample-based discretization method. It is employed to down-sample an input design (input image, hidden-layers, output matrix, etc.), and compressing it is dimensionality and enabling assumptions about the components available in the binned sub-regions to be made. This will decrease the size of learning parameters and provide fundamental interpretation invariance to internal depiction, thus further reducing the cost of computation. Our model adopted the kernel size of $3 \times 3$ during the Maxpooling process. After the final convolution block, the network flattened to one dimension.

**3.2.4 Batch normalization.**   Batch normalization enables every layer of the network to learn a little more independently of the other layers. It also normalizes the output from the previous activation layer by subtracting the batch mean and dividing the batch standard deviation [41] to improve the steadiness of the neural network.

**3.2.5 Fully connected layer.**   This layer takes the output of the previous layer (Convolutional, ReLU, or Pooling) as its input and calculates the probability values for classification into the various groups.

**3.2.6 Loss function.**   This layer applies a soft-max function to the input data sample. This layer is used for the final prediction. Therefore, our loss function is given as:

$$L_i = -\log\left(\frac{e^{\beta_y}}{\sum_j^c e^{\beta_j}}\right)$$

(7)

Where $\beta_j$ is the *jth* element of the vector of class scores $\beta$, $\beta_y$ is the CNN score for the positive class and *c* is classes for each image. The softmax ensures a proper prediction probability in the log of the equation.

**3.2.7 Regularization.**   An efficient regularization method named as a dropout is employed. This strategy was being proposed by Srivastava et al. [42]. During the training process, the dropout is conducted by maintaining the neuron active with a certain probability *P* or by setting it to 0. In our study, we set hyperparameter to 0.50 because it outputs in the maximum amount of regularization [43].

## 3.3 Transfer learning

Knowledge transfer from source to target tasks is often the only option in highly technical domains in which the availability of huge-scale quality data tends to be challenging. The use of pre-trained weights is not only an effective optimization technique but often supports

classification sensitivity. The first layer of CNN learns to identify common characteristics such as borders, textures, or patterns. In contrast, the upper layers concentrate more on sophisticated and detailed aspects of the image, such as pathological lesions. Training only the top layer of the network with the target dataset while adopting the remaining layers' initialized parameters is the widely used method, especially in the computer-aided diagnosis (CAD) domain. Apart from performance benefits, fewer training parameters often reduce the chance of over-fitting, which is a severe issue in the training cycle for Neural Networks [44]. A brief overview of the transfer learning-based CNNs used for automatic COVID-19 detection given in this section. This study used the Keras DL Framework by F. Chollet. [45] and TensorFlow as backend, which includes pre-trained, DL models made available in Keras Applications alongside weights. In this analysis, we employed VGG16, InceptionV3, and Xception model with pre-trained weights on the ImageNet database as S. Pal. [46], and K. Simonyan [47]. Based on WorldNet's hierarchical structure, the ImageNet data comprises over 3.2 million perfectly annotated images, spread over 5247 categories [48]. Hence, the employed models are explained below.

**3.3.1 VGG16.** The pre-trained VGG16 framework has learned to obtain image characteristics that can differentiate between one image class to another and displayed good results when applied to image classification and recognition in other target tasks [46, 47]. The VGG16 Deep CNN model or VGGNet, which includes 144 million parameters, comprises 16 convolution layers with a deep visual field of $3 \times 3$, five max-pooling layers of size $2 \times 2$ for spatial pooling, followed by three fully connected layers with the final layer as the soft-max layer [48]. ReLu applied to all hidden layers. The model also uses the regularisation of the dropouts in the fully connected layers. If the fully connected classifier (or bottleneck layer) eliminated from the pre-trained VGG16 network, it could be used as a feature vector generator for our images to generate semantic image vectors. VGG16 model used as pre-trained and softmax as a classifier. The general structure and parameters of VGG16, which yielded high-performance accuracy in chest x-ray images, are presented in Table 2 and hyper parameters are presented in Table 3.

**3.3.2 InceptionV3.** Szegedy et al. [49] first introduced the "Inception" micro-architecture in their paper. The inception module aims to serve as a "multi-level function extractor" by

Table 2. The layers and layer parameters of the VGG16 model.

| Layers | layer Type | Output Shape | Trainable parameters |
|---|---|---|---|
| 1 | Cov2d | [224, 224, 64] | 1792 |
| 2 | Cov2d | [224, 224, 64] | 36928 |
| 4 | Cov2d | [112, 112, 128] | 73856 |
| 5 | Cov2d | [112, 112, 128] | 147585 |
| 6 | Cov2d | [56, 56, 256] | 295168 |
| 7 | Cov2d | [56, 56, 256] | 590080 |
| 8 | Cov2d | [56, 56, 256] | 590080 |
| 9 | Cov2d | [56, 56, 256] | 590080 |
| 10 | Cov2d | [28, 28, 512] | 1180160 |
| 11 | Cov2d | [28, 28, 512] | 2359808 |
| 12 | Cov2d | [28, 28, 512] | 2359808 |
| 13 | Cov2d | [14, 14, 512] | 2359808 |
| 14 | Cov2d | [14, 14, 512] | 2359808 |
| 15 | Cov2d | [14, 14, 512] | 2359808 |
| 16 | Cov2d | [14, 14, 512] | 2359808 |

**Table 3. Hyper-parameters of the VGG16 model and preferred weights in this study.**

| R1 | R2 | R3 | R4 | R5 | R6 | R7 | R8 |
|---|---|---|---|---|---|---|---|
| **VGG16** | 224*224 | ADAM | 32 | 10-fold | 1e-4 | BCE CCE | 20 |

R1—Model, R2—Image Size, R3—Optimizers, R4—Mini Batch Size, R5—cross validation, R6—Initial Learning Rate, R7—Loss function, R8—Epoch, BCE—Binary cross-entropy, CCE—Categorical cross-entropy.

computing $1 \times 1$, $3 \times 3$, and $5 \times 5$ convolution layers within the same network module—these filters' output is then stacked along the channel dimension and then fed into the next network layer. The Inception V3 framework present in the Keras core obtained from the later released version by Szegedy et al. [50] in "Rethinking the Inception Framework for Computer Vision (2015)". The Inception-v3 model trained for ImageNet benchmark datasets with images from 1000 label classes [51]. This model comprises two parts: feature extraction part with a CNN and classification part with fully connected and softmax layers. Selected hyper parameters are shown in Table 4.

**3.3.3 Xception.** F. Chollet. [45] introduced a CNN neural network that entirely focused on depthwise separable convolution layers known as Xception. This architecture has 36 convolutional layers that form the network's base for extracting features. The 36 convolutional layers made up of 14 modules, all of which have residual linear associations surrounding them, except the first and last modules. Under the MIT license, an open-source Xception implementation using Keras and TensorFlow supported the Keras Applications module. This architecture provides better performance than Inception V3, ResNet-50, ResNet-101, ResNet 152, and VGG on ImageNet dataset. We preferred these models because they are well developed and have shown excellent results when applied to a variety of classification cases for medical images. Hyper parameters selected for this model are shown in Table 5.

## 3.4 Proposed method using transfer learning

The developed scheme encompasses two main components: image pre-processing and the second is a deep transfer framework. Fig 5 shows the proposed pre-processing/ Deep Transfer Learning Model. The morphological operation was mainly used for the pre-processing step,

**Table 4. Hyper-parameters of the InceptionV3 model and preferred weights in this study.**

| R1 | R2 | R3 | R4 | R5 | R6 | R7 | R8 |
|---|---|---|---|---|---|---|---|
| **InceptionV3** | 224*224 | ADAM | 32 | 10-fold | 1e-4 | BCE CCE | 20 |

R1—Model, R2—Image Size, R3—Optimizers, R4—Mini Batch Size, R5—cross validation, R6—Initial Learning Rate, R7—Loss function, R8—Epoch, BCE—Binary cross-entropy, CCE—Categorical cross-entropy.

**Table 5. Hyper-parameters of the Xception model and preferred weights in this study.**

| R1 | R2 | R3 | R4 | R5 | R6 | R7 | R8 |
|---|---|---|---|---|---|---|---|
| **Xception** | 224*224 | ADAM | 32 | 10-fold | 1e-4 | BCE CCE | 20 |

R1—Model, R2—Image Size, R3—Optimizers, R4—Mini Batch Size, R5—cross validation, R6—Initial Learning Rate, R7—Loss function, R8—Epoch, BCE—Binary cross-entropy, CCE—Categorical cross-entropy.

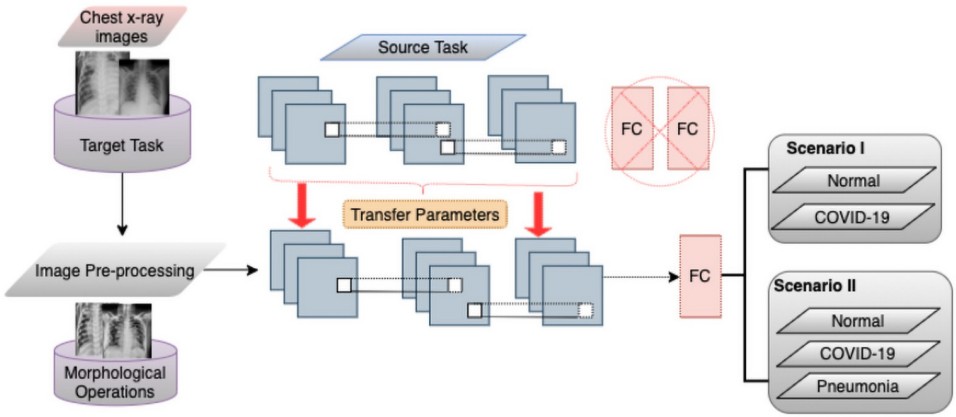

**Fig 5. Proposed pre-trained method for COVID-19 detection.**

whereas the deep transfer framework was used during the training, validation, and testing phase. The proposed model in details, Set of transfer learning models be defined by $\Gamma = \{VGG16, InceptionV3, Xception\}$. The top layer of each deep transfer model is removed and trained with the COVID-19 X-ray Images dataset (X, Y); where X the set of N input data, each of size, $224 \times 224$, and Y have the same class, $Y = \{y/y\epsilon\{Normal; COVID-19; Pneumonia\}\}$. Dataset segregated to training and testing sets, training set ($train_x$; $train_y$) by 80 percent for training, and 20 percent for testing. The selection of 80 percent for training and 20 percent for validation has proven effective in many research areas. The training samples is then divided into mini-batches, each of size $n = 32$, so that each mini-batch $(X_i; Y_i)\epsilon(train_x; train_y)$ where, $i = 1, 2, 3, \ldots\ldots, \frac{N}{n}$ and iterative process optimises the CNN framework for the reduction of functional loss as demonstrated in Eqs 7 and 8 for categorical and binary cross-entropy loss.

$$L_b = -\sum_{i=1}^{c'=2} t_i log(\beta_i) = -t_1 log(\beta_1) - (1 - t_1) log(1 - \beta_1) \qquad (8)$$

where, $C'$ two classes $c1$ and $c2$; $t[0, 1]$ and $\beta_1$ for class $c1$; $t_2 = 1 - t_1$ and $\beta_2 = 1 - \beta_1$ are the ground-truth and the score for $c2$.

## 3.5 Experiment setup

We divided our classification tasks in the scenario I and scenario II as represented in Table 6. The respected classes were train in our built CNN and 3 pre-trained models; *VGG16*, *InceptionV3*, and *Xception* respectively. The image resolution has standardized to a uniform size following the input requirements of each model. The number of epochs, i.e., completely forward and backward passes through the network set 20 to the already pre-trained models and 50 for build CNN. The training/testing split set to 80/20. The stratified random sampling performed

**Table 6. Description of classification task.**

| Scenario | Classification |
|---|---|
| I | Normal / COVID-19 |
| II | Normal / COVID-19 / PNEUMONIA |

to ensure proportional class distribution. Mini-batch size set to 32, and the binary cross-entropy and categorical cross-entropy loss function selected due to its suitability for a binary-class classification and multi-class classification task. The default Optimiser for pre-trained models was Adam with a learning rate of $1e-4$, and for the build, CNN is RMSprop with a learning rate of $3e-4$. The standard evaluation metric of Accuracy (Acc), Sensitivity (Se), and Specificity (Sp) on testing data set used for final results validation.

## 3.6 Performance improvement

**3.6.1 Fine-tuning for pre-trained models.** The classification algorithms adopted for the study were pre-trained on a large-scale collection of ImageNet data covering various classes such as cars, fruits, horses, etc. The algorithms achieve superior performance for the object-based training data set on classification tasks, while proving restricted in their deployment to narrow domains, such as medical lesion detection. Detection of abnormal symptoms in chest x-ray images depends on a broad range of specific features within an image and their positions. There is a new representation of input image in each CNN layer by progressive extraction of the most distinctive features. For instance, the first layer of a network can learn edges, for example, while the last layer can identify lung opacification for COVID-19 and Pneumonia classification. Therefore, the following parameters considered in this experimental study: (i) the removal of the top layer and the re-training of the network; and our adopted approach (ii) the removal of the *n* top layers and the re-training. The parameter *n* varies around the utilized CNNs, which depends on the total number of layers in each model structure. The *n threshold* was selected, and the following segments of the model were 'un-frozen' and fine-tuned. The preliminary *n* layers were regarded as a fixed-feature extractor [52], while the subsequent layers fitted to unique x-ray characteristics.

**3.6.2 Data re-sampling.** In a classification task, a challenge of unbalanced classes can observe when the data set has a meager number of sample data in one or more types, which leads to the problem of misclassification. Data set can be balanced using under-sampling or over-sampling methods. In this study, we implemented both the processes in the dataset. Undersampling performed by randomly deleting the classes with sufficient datasets and over-sampling performed using Keras DL library via the class *ImageDataGenerator* class.

**3.6.3 Contrast enhancement.** To accomplish contrast enhancement in the chest x-ray images, mathematical morphology has been used. Mathematical morphology approaches work hinged on the structural values of objects. To pull out the elements of an image, these methods use relationships between classes and mathematical fundamentals, which help explain areas. In morphological operators, the input consists of two data sets. The original image is included in the first set, and the second one illustrates the structural element (SE), which is also called a mask. The original image is in grey level or binary, and the mask is a $0s$ and $1s$ value matrix [53]. In morphological operators, if the gray-level image matrix id represented by $I(x, y)$ and the SE by $S(u, v)$, the erosion and dilation operators are defined as Eqs 9 and 10 [54].

$$I \ominus S = \min_{u,v}\{I(x + u, y + v) - S(u, v)\} \tag{9}$$

$$I \oplus S = \max_{u,v}\{I(x - u, y - v) + S(u, v)\} \tag{10}$$

The erosion operator decreases the objects' size and increases the size of an image's holes and eliminates very tiny information from that image. It makes the final image appear darker than the original image by removing bright areas under the SE. The dilation operator operates

in reverse; in other words, the size of objects increases and holes in the image decreases, respectively. Therefore, the opening operator is similar to implementing the dilation and erosion operations on the same image Eq 11, while the closing operator acts in reverse Eq 12.

$$I \circ S = (I \Theta S) \oplus S \tag{11}$$

$$I \bullet S = (I \oplus S) \Theta S \tag{12}$$

The opening operator eliminates poor relations between artifacts and small information, while small gaps are eliminated, and the closing operator fills cracks. The size and shape of a SE are usually chosen arbitrarily; however, disk-shaped SE is used more frequently than other masks for medical images.

The top-hat result is obtained from the variation between the source images and its opening with a SE. In contrast, the variation between the closure by a SE and the input image defines the bottom-hat transform output. In the case of the top-hat transform, bright objects that are shorter than the SE fetched. On the other side, dark components obtained with the transformation of the bottom hat, which is inferior to the SE. Thus, by integrating the additional outcome of top-hat and the subtraction result of bottom-hat with its original image, we can create an improved image in which the essential objects are more visualized. Fig 6 illustrate top-hat performance, Bottom-hat, and the product of combined transform. The equation can assess with top-hat, bottom-hat, and enhanced picture where $I$ is the input image, then $S$ is the structuring part, $\circ$ means to opening and $\bullet$ to closing.

$$I_{th} = I - (I \circ S) \tag{13}$$

$$I_{bh} = (I \bullet S) - I \tag{14}$$

$$I_{comb} = I + I_{th} - I_{bh} \tag{15}$$

## 3.7 Visualize feature maps

The feature maps, or activation maps, record the input applied with filters, such as the source images or other feature maps. The purpose of visualizing a feature map for particular source images would explain what attributes in the feature maps are observed or retained. The idea would be that the feature maps near the input detect fine-grained or small information while featuring maps near the model output to capture more distinctive characteristics. The first layer of CNN always learns features like edges, lines patterns, color, and deeper layer network to identify more complex features like pathological lesions. Later layers construct their features by merging features from previous layers. To analyze the visualization of feature maps, we used the highest performed model with x-ray, i.e., VGG16 model, and used to create activations. The activations for VGG16 network models shown in Fig 7.

## 3.8 Classification performance analysis

Different metrics have been used to evaluate the efficiency of the highest performing DL model in the Scenario I and Scenario II. To calculate the true or false classification of the COVID-19 diagnosed in the Xray images evaluated as follows. Initially, the cross-validation estimator [55] is adopted and plotted in a confusion matrix as shown in Table 7. The confusion matrix has the following four predicted outcomes. True Positive (TP) has been identified with the right diagnosis and a variety of abnormalities. True Negative (TN) is an erroneously calculated number of periodic instances. False positives (FP) are a set of periodic instances. The

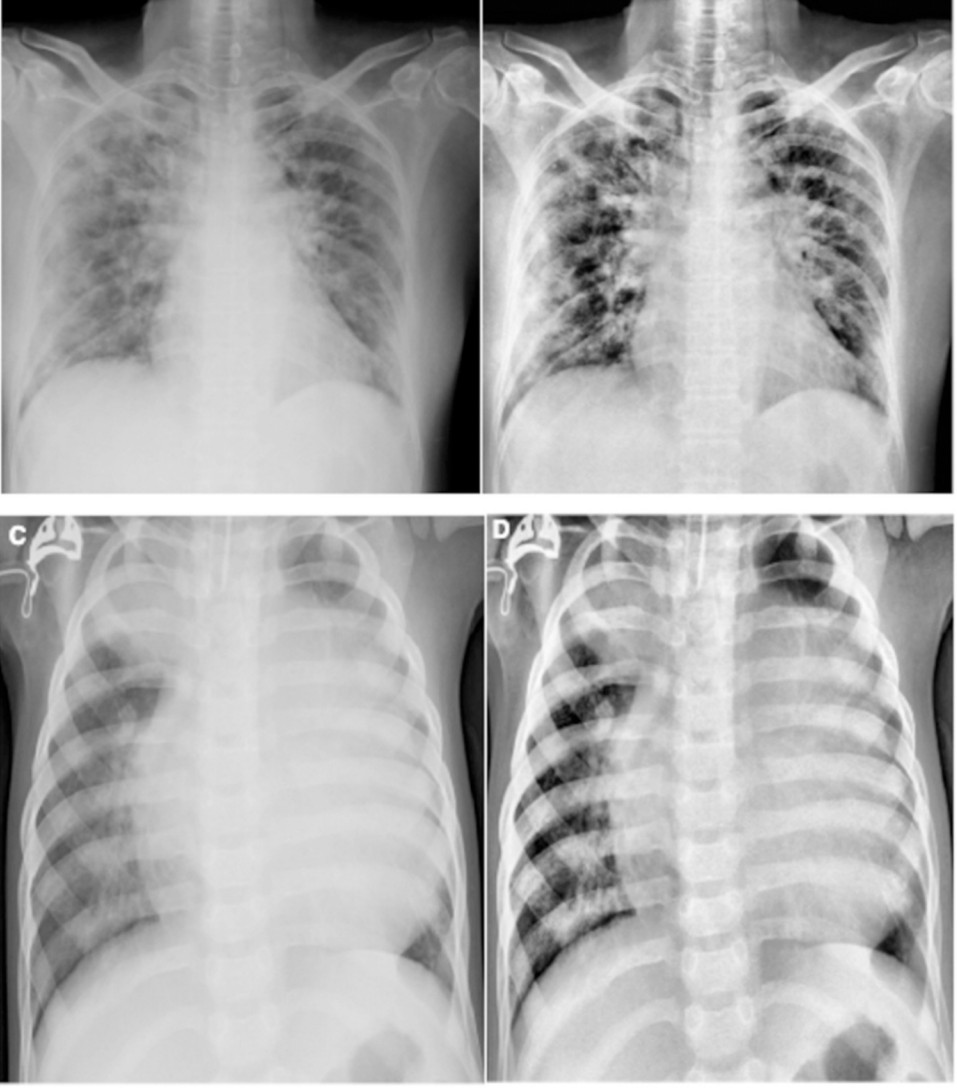

**Fig 6. (A) and (C) original chest x-ray COVID-19 and Pneumonia images; (B) and (D) contrast enhanced image using $I_{comb}$.**

following performance metrics are used to calculate the values of possible outcomes in the confusion matrix Table 7.

*Accuracy*: Accuracy is an essential metric for the evaluation of the results of DL classifiers. It is a summary of the true positive and true negatives divided by the confusion of the matrix components' total values. The most accurate model is an excellent one, but it is imperative to ensure that symmetric sets of data with almost equal false positive values and false negative values. Thus, the elements of the confusion matrix mentioned above will be calculated to evaluate the effectiveness of our proposed classification model for the COVID-19 database.

$$Accuracy(\%) = \frac{TP + TN}{TP + FN + TN + FP}100\%. \qquad (16)$$

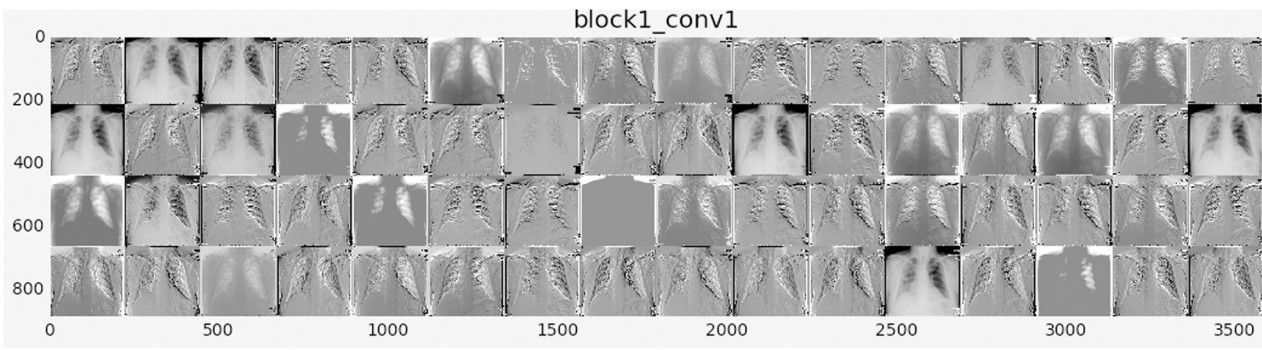

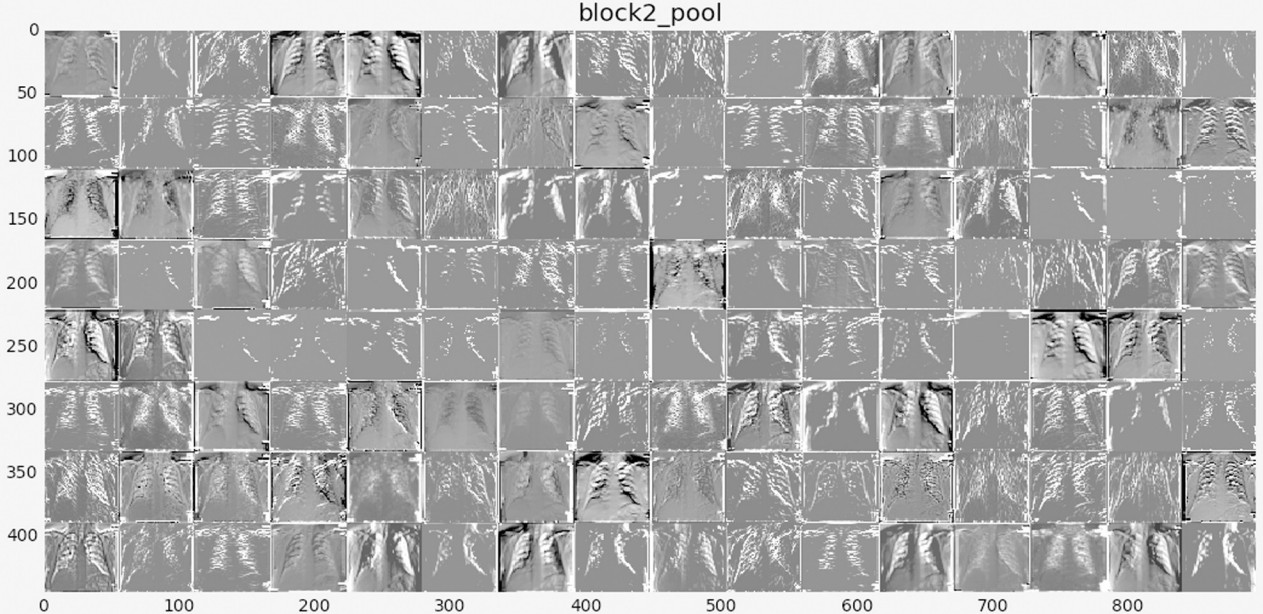

**Fig 7. Visual feature maps in first layer and deep layer.**

*Sensitivity (Recall)*: sensitivity is measured as the number of accurate positive predictions divided by the sum of positive. The best sensitivity is 1.0, whereas the worst is 0.0. We calculate sensitivity using following equation;

$$Sensitivity = \frac{TP}{TP + FN} \tag{17}$$

**Table 7. Confusion matrix.**

|  | Predictive Positive | Predictive Negative | Total |
|---|---|---|---|
| Actual Positive | TP | FN | TP + FN |
| Actual Negative | FP | TN | FP + TN |
| Total | TP + FP | FN + TN |  |

TP = True Positive, FN = False Negative, FP = False Positive, TN = True Negative.

*Specificity*: specificity is measured as the number of correct negative predictions divided by the sum of negatives. The best specificity is 1.0, whereas the worst is 0.0. We calculate sensitivity using the following equation;

$$Specificity = \frac{TN}{TN + FP} \tag{18}$$

*Precision*: precision is the fraction of the correct positive labelled by our model to all negative labelled. Precision has been calculated as follows;

$$Precision = \frac{TP}{TP + FP} \tag{19}$$

## 4 Experiments results

The algorithms implemented using MatLab for morphological operations, Keras 2.3 library, with TensorFlow 2 as a back-end and Python 3.8 programming language in jupyter notebook with a processor of 2.3 GHz Intel Core i9 and RAM of 16 GB 2400 MHz DDR4 with Intel UHD Graphics 630 1536 MB.

### 4.1 Results by selected models

The implemented models are tested in two distinct scenarios. The proposed CNN and 3 pre-trained CNN models are evaluated to obtain accuracy on the test data set. Testing accuracy (%) is used to estimate the accuracy and precision of the proposed model, which is presented in Tables 8 and 9. The confusion matrix is also one of the precise metrics that further insight into the test accuracy. At first, a scenario I data was trained with three pre-trained models, and the highest performance model is selected; after that proposed CNN model was trained with the same data, and confusion matrices present the result in Fig 8. Similarly Fig 9 summarizes the confusion matrices for scenario II. However, in pre-trained models, we additionally

**Table 8. Average performance of the pre-trained CNN models.**

| Scenario | Classes | Model | Results* | Results** |
|---|---|---|---|---|
| I | Normal/COVID-19 | *Xception* | *Acc* = 81.82%, *Se* = 72%, *Sp* = 100% | *Acc* = 89.09%, *Se* = 81%, *Sp* = 100% |
| | | **VGG16** | *Acc* = 97.62%, *Se* = 95%, *Sp* = 100% | **Acc = 100%, Se = 100%, Sp = 100%** |
| | | *InceptionV3* | *Acc* = 96.49%, *Se* = 100%, *Sp* = 93.33% | *Acc* = 96.49%, *Se* = 100%, *Sp* = 93.33% |
| II | Normal/COVID-19/PNEUMONIA | *Xception* | *Acc* = 68%, *Se* = 86.36%, *Sp* = 100% | *Acc* = 73%, *Se* = 75%, *Sp* = 100% |
| | | **VGG16** | *Acc* = 83.50%, *Se* = 94.43%, *Sp* = 100% | **Acc = 87.50%, Se = 96.43%, Se = 100%** |
| | | *InceptionV3* | *Acc* = 83%, *Se* = 100%, *Sp* = 100% | *Acc* = 86%, *Se* = 100%, *Sp* = 100% |

Results* = Results obtain before model fine-tune and contrast enhancement in x-ray images, Results** = Results obtain after model fine-tune and contrast enhancement in x-ray images, *Acc* = Accuracy, *Se* = Sensitivity, *Sp* = Specificity.

**Table 9. Average performance of the build CNN models.**

| Scenario | Classes | Model | Results |
|---|---|---|---|
| I | Normal/COVID-19 | *CNN* | **Acc = 97.67%, Se = 100%, Sp = 95.45%** |
| II | Normal/COVID-19/PNEUMONIA | *CNN* | **Acc = 93.75%, Se = 100%, Sp = 95.24%** |

Results = Results obtain after model fine-tune and contrast enhancement in x-ray images, *Acc* = Accuracy, *Se* = Sensitivity, *Sp* = Specificity.

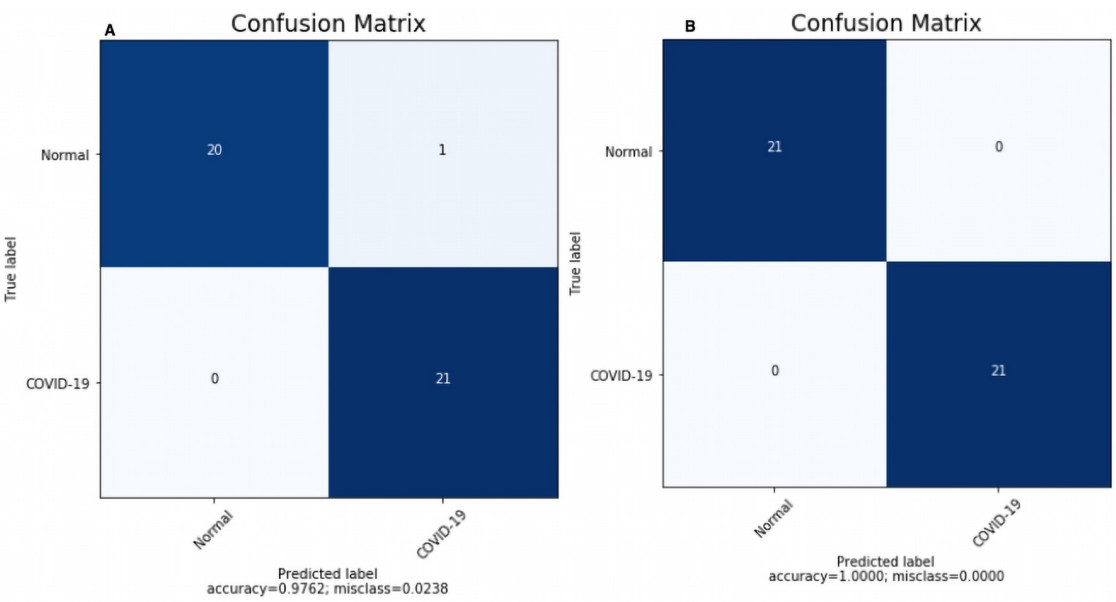

**Fig 8. Confusion matrices for scenario I obtained by (A) build CNN and (B) VGG16.**

applied fine-tuning as a complement to the default setting. The results obtained for each model used for comparison purposes after the exclusion and retraining of n layers (depending on the model layers number) as represented in Table 8. The fine-tuning impact is measured as a percentage increase/decrease in accuracy. The best accuracy is selected for each model (default or fine-tuning). Eventually, the VGG16 reported as a top 1 CNN architectures with the highest classification efficiency on x-ray data in both the scenarios, as shown in Table 8. The accuracy was further plotted after each epoch to examine the models' convergence capabilities in the fine-tuning scenarios.

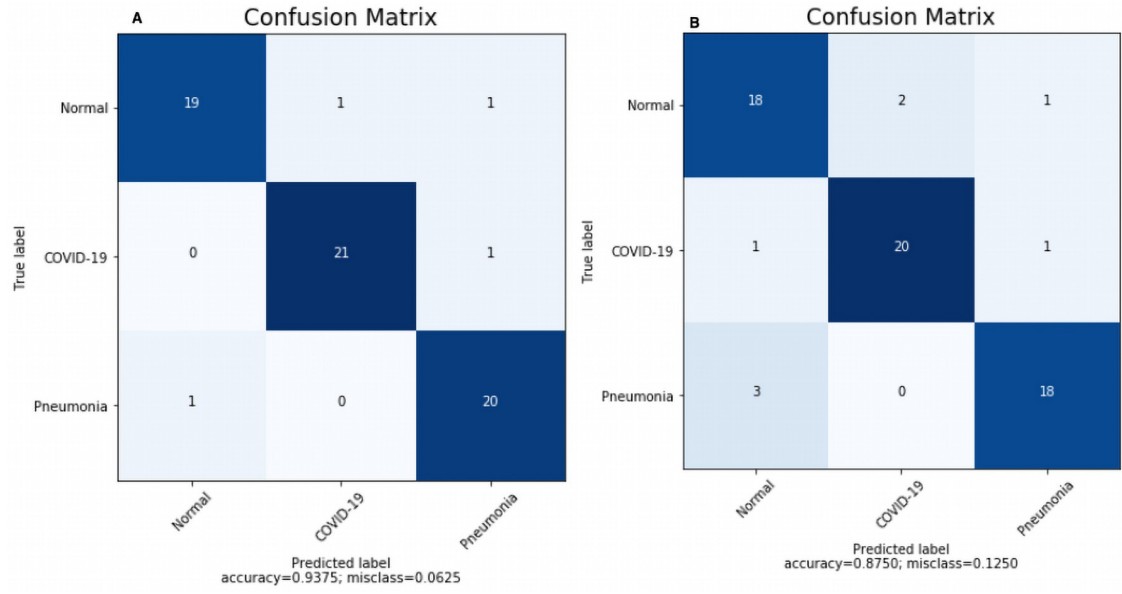

**Fig 9. Confusion matrices for scenario II obtained by (A) build CNN and (B) VGG16.**

## 4.2 Explaining proposed model using Grad-CAM

To make deep learning more practical and explainable, a range of work was performed. It is also essential to make the deep neural network more interpretative in various deep learning applications linked to medical imaging. A technique of Gradient Weighted Class Activation Mapping (Grad-CAM) is developed by Selvaraju et al. [56], which provides an illustrative view of deep learning techniques. The technique of Grad-CAM offers a visual description for any deeply related neural network. This helps to decide more about the model when conducting identification or prediction tasks. The simple chest x-ray image is given as input and uses the proposed model as a detection method. After calculating the predicted label using the full model, Grad-CAM is applied to the last Convolution layer. Fig 10 shows the heatmap visualization on chest x-ray images by the proposed model.

## 4.3 Improvement techniques comparison

Three pre-trained models and proposed CNN frameworks are employed considering the data limitation in current work. Another issue observed was data imbalance, causing over-fitting and poor generalization to the test data on classification accuracy. Therefore, *ImageDataGenerator* is employed in pre-trained models to avoid such issues. Image data-enhanced using contrast enhancement and removed underexposed and overexposed images. Thus, Table 8 *Results*\* summarizes the results before fine-tune with contrast enhancement. Similarly, Table 8 *Results*\*\* summarizes the results after fine-tuning with contrast enhancement. In this work, the pre-trained *VGG16* model yielded high-performance accuracy in both scenarios. Fig 11 compares the highest accuracy obtained for both the scenarios and Fig 12 summarize the accuracy obtained for both the scenarios. It is observed that image enhancement and fine-tunes contribute to performance improvement. Meanwhile, new build CNN employed a cross-validation ($K = 10$) re-sampling procedure along with *ImageDataGenerator* to evaluate the new CNN model on a limited x-ray data sample. Table 9 represents the results obtained by the new build CNN in both the scenarios.

Confusion Matrix for the scenario I is compared in Fig 8(A) using proposed CNN, and Fig 8(B) using proposed VGG16. For the scenario I Fig 8(A), all the cases classified correctly except one COVID-19 image as healthy; otherwise, this model would have reach 100% accuracy. Similarly, in Fig 8(B) all the cases classified correctly with 100% accuracy. Confusion Matrix for scenario II compared in Fig 9(A) using proposed CNN and Fig 9(B) using VGG16. However, interestingly proposed CNN in multi-class classification performance was better than the multi-class classification of VGG16, although the classes were miss-classified in both scenarios. Finally, the Receiver Operating Characteristics (ROC) Curves applied to check the classification efficiency of the highest performing classifier by revealing the true positive rate (TPR) against the false positive rate (FPR) to detect true positive of COVID-19 in the x-ray images evaluated in Figs 13 and 14.

# 5 Discussion

## 5.1 Automated COVID-19 detection

Based on the research findings, we observed that proposed CNN and pre-trained CNN models do have major impacts on the automatic identification and extraction of essential features from chest x-ray images relevant to the detection of COVID-19. Cohen et al. [26] gathered radiology COVID-19 images from multiple sources for study to establish a precise model for the effective diagnosis of this viral infection. Several of the studies listed in this section of the article used Cohen et al. [26] COVID-19 image data, and for other instances, such as Non-

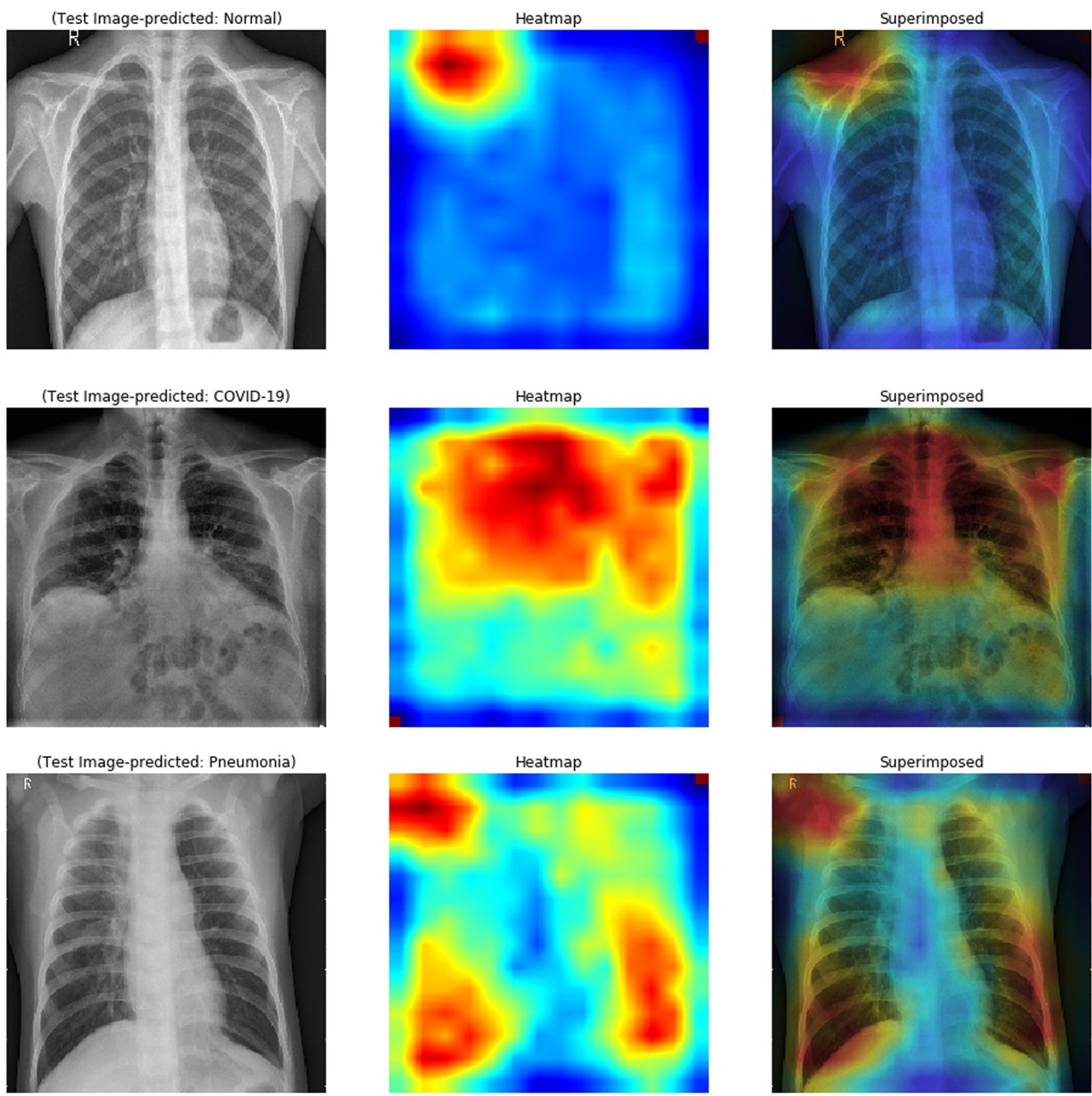

**Fig 10. Visualisation on chest x-ray of normal/COVID-19/pneumonia infected using Grad-CAM on the proposed model.**

COVID-19, images collected from different sources that are publicly accessible Kermany et al. [39]. Ozuturk et al. [10] proposed DarkCovidNet for identifying COVID-19 using chest x-ray images and obtained the accuracy of 98.08% in the classification of Normal and COVID-19 and 87.0% accuracy in Normal, COVID-19 and Pneumonia classification. Similarly, COV-IDX-Net was proposed by Hemdan et al. [57] for diagnosing COVID-19 using x-ray images. Their work used 25 positive COVID-19 and 25 normal images and successfully acquired 90% accuracy. On the other hand, Wang et al. [9] developed COVID-Net, a DL model, for the detection of COVID-19. They obtained an accuracy of 92.4% using a sample of 16,756

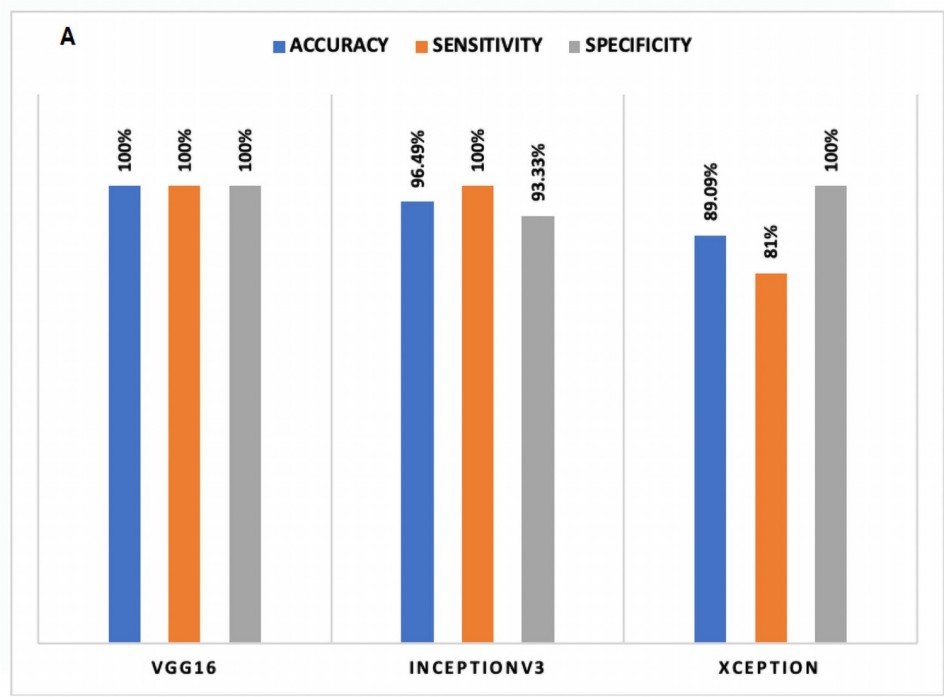

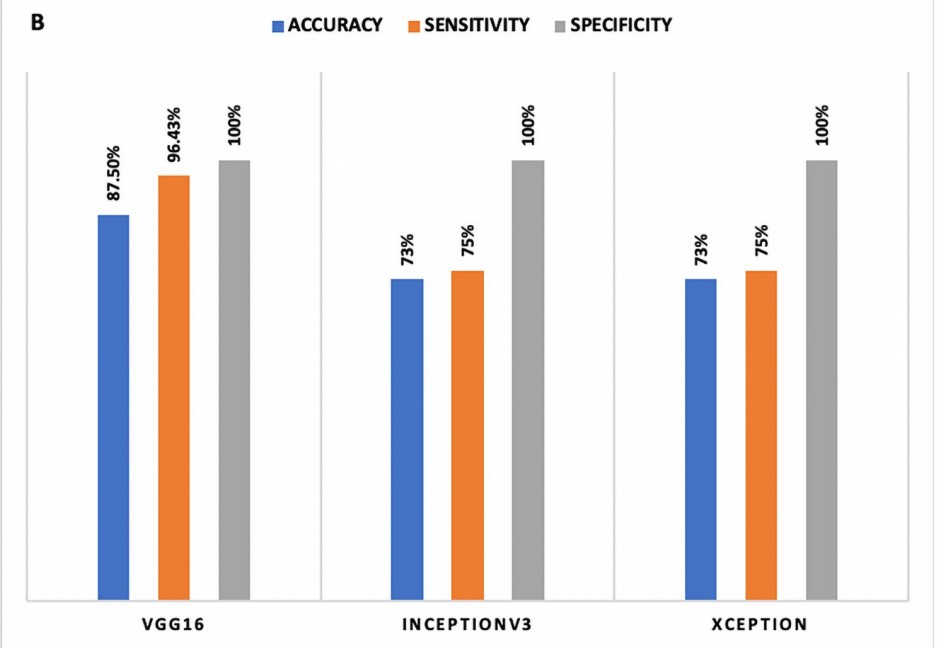

**Fig 11. Comparison of accuracy achieved in selected models after fine-tune: (A) scenario I and (B) scenario II.**

radiographic images from various open-source data. The concept of knowledge transfer adopted by Ioannis et al. [58] for a similar objective as COVID-Net. They used 224 confirmed COVID-19, 700 pneumonia, and 504 normal radiology images in their study and achieved a 98.75% performance for the binary class and 93.48% for the multi-class classification. We

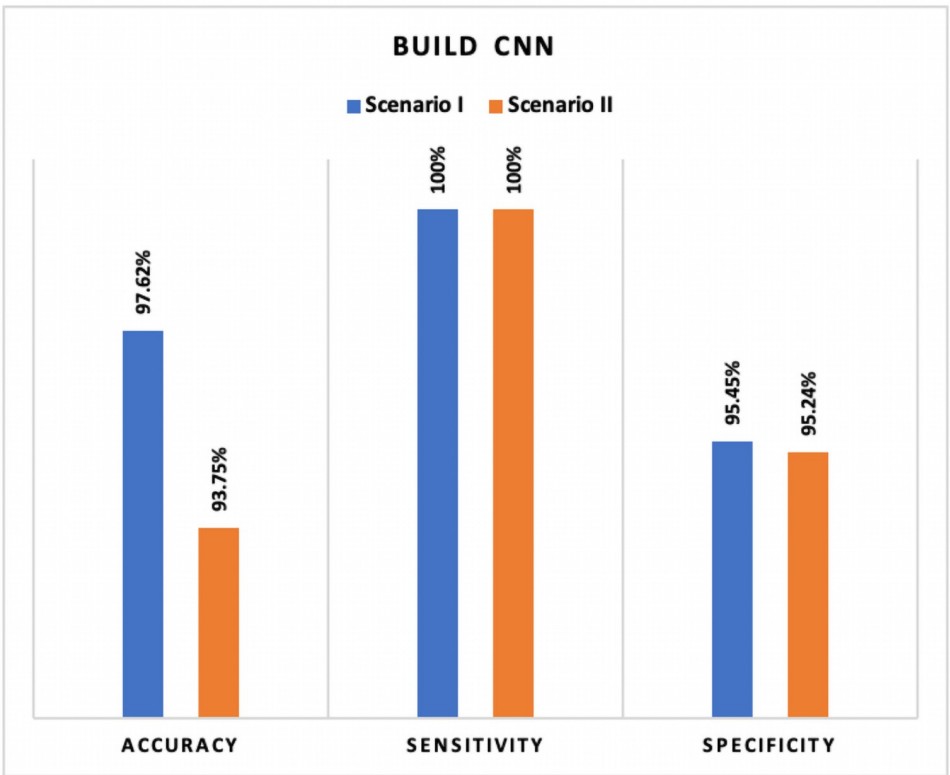

**Fig 12. Comparison of accuracy achieved in new CNN: Scenario I and scenario II.**

noticed an increase in the number of classification categories, the performance accuracy of the CNN based model decreased. Author Sethy et al. [59] combined DL and ML techniques to develop a system. In their study, they used the ResNet50 model to retrieve image features and SMV as a classifier. Using 50 images, they achieved 95.38% accuracy. Several researchers used CT images to train their CNN based models and achieved less accuracy than the researcher who used x-ray images. For instance, Ying et al. [60] obtained an accuracy of 86% utilizing CT images with a ResNet50, called DRE-Net. Wang et al. [61] modified Inception deep model developed a system using CT images; they successfully obtained a classification accuracy of 82.9%. Zheng et al. [62] introduced a 3D CNN model for detecting COVID-19 using CT

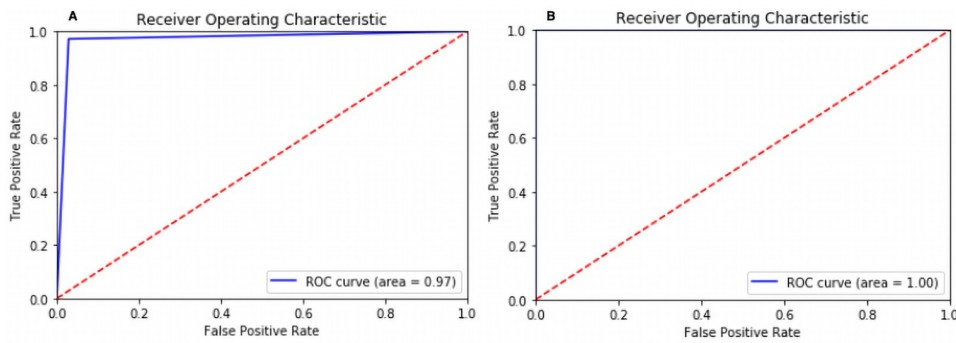

**Fig 13. ROC curve in scenario I for (A) build CNN and (B) VGG16.**

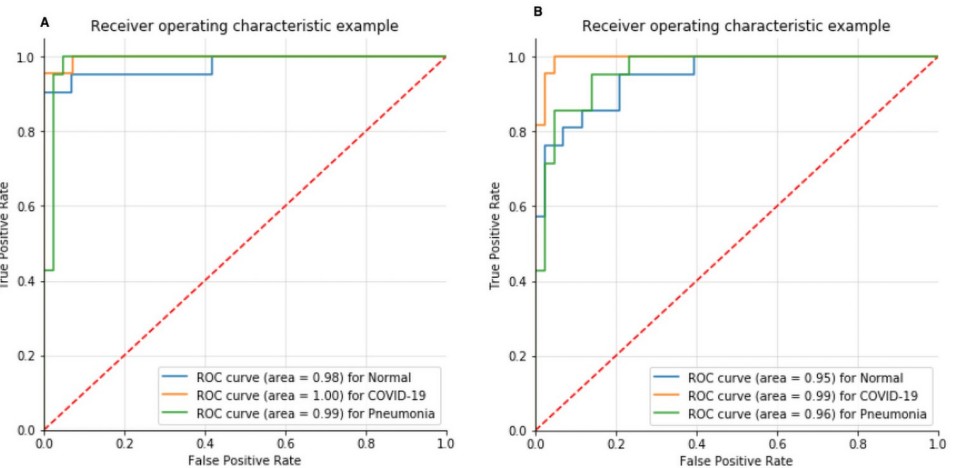

**Fig 14. ROC curve in scenario II for (A) build CNN and (B) VGG16.**

images and obtained an accuracy of 90.8%. Xu et al. [18] employed ResNet using CT images and achieved an accuracy of 86.7%. Several of the previous studies utilized little data to generate a COVID-19 detection system.

## 5.2 Performance improvement

The initial phase of the experiment involved the extraction of features initialized using the pre-trained CNN models through transfer learning and subsequent removal of the top layer (existing approach). A detailed evaluation of 3 convolutional networks (state-of-the-art) conducted. In the second phase, the *n* layers were unfreezing' and progressively re-trained to adequately conform to the details of the case study (adopted approach) method. The integration of [26, 39] data sets was performed to create classes for system training. The scale of the data used in training significantly impacts the outcome of the method of neural networks [63]. The various pre-processing measures were implemented in x-ray images to improve classification accuracy. Medical images prove extremely challenging to differentiate pathological lesions; therefore, image enhancement used to enhance lesions' visibility (e.g., contrast enhancement). The high accuracy achieved in the VGG16 model with, yielding 100% for binary classification and 87.50% for multi-class classification. The lowest performance was acquired by Xception 81.82% for binary and 68% for multi-class classification. The fine-tuning effect (un-freezing layers) varied across the models. Improvement in accuracy was only slight but efficient enough to make a difference. There are seven different optimizers for CNN architecture. In this experiment, we used the ADAM optimizer as a default in which VGG16 + Adam produced outstanding performance accuracy for COVID-19 detection. CNN based transfer learning has typically produced promising results in computer vision, but still, as far as classification performance is concerned, these architectures are less suitable for multi-class classification of medical images. Study findings have shown that existing deep learning algorithms are unsuccessful in classifying small data sets of multi-class images. As classes multiplied, the performance of pre-trained deep learning techniques was dropped. This outcome, in reality, is very normal. It is relatively common to have better results in two categories classification (100 percent) than three-category classification (87.50 percent). As classes multiplied, the expected accuracy at normal sample dropped. This finding was consistent with previous research [64]. Therefore, our main goal is to build a deep learning based model which not only perform binary classification efficiently

but also gives efficient performance in multi-class classification of chest infection diseases. However, the application of pre-trained deep learning in medical practice still poses many challenges. Thus, we trained CNN architecture that can be trained from scratch on x-ray images, works as a standardized architecture, and subsequently generates better classification accuracy for multi-class classification. Therefore, we build a new CNN which yielded accuracy 97.62% in binary classification and 93.75% in multi-class classification Table 9.

## 5.3 Approach limitation

Limitations of the study can be summarized as follows; Initially, due to the limited availability of high quality COVID-19 public image, only a small-to-moderate data set size used in the study. The literature review suggested for this research is at its initial stage, because most of the existing AI systems introduced in the research literature have been closed-source and inaccessible for the scientific community to develop a better understanding and expansion of these systems. DL models pre-trained on an ImageNet dataset used as a compensation procedure due to the limited number of datasets. Several experiments performed with different parameters to achieve high accuracy. However, the history of DL architectures is not well known in many situations and is regarded as a black box. Thus, the proper structure and optimum values for the number of layers and nodes in different layers are still challenging. The selection of values for the learning rate, number of epochs, and regularizer intensity often involve unique domain knowledge. In this study, a pre-trained VGG16 network with defined hyperparameters yielded the highest performance accuracy than other tested systems in binary classification and achieved.

Similarly, a newly developed model with defined layers and hyperparameter yielded the highest accuracy in multi-class classification. Although with certain limitations, we believe our system can be advantageous to assist experts in identifying COVID-19 infection using x-ray images. X-ray scans were recommended in this study because they are widely available for the diagnosis of disease.

## 6 Conclusion and future work

The COVID-19 pandemic remains a serious threat that has caused chaos around the globe. This epidemic continues to pose a threat to personal health in various ways around the world, including mortality. Early detection of COVID-19 in a patient can reduce mortality and save lives. In this research, we introduced a methodology focused on DL to classify and detect the COVID-19 cases from x-ray images. Our model is entirely automated and is capable of categorizing binary class with 100% accuracy using VGG16 and multi-class with 93.75% using a built CNN. Accuracy obtained by existing models and models used in this study is shown in Table 10. The study used the limited sets of data from diverse sources to analyze system robustness through its ability to respond to real-world scenarios. This framework effectively operates as an additional screening tool for COVID-19 detection. The proposed models can address a shortage of radiologists in rural areas and used to classify chest-related diseases such as viral pneumonia and COVID-19. The system implemented is fully prepared for testing with a considerably larger directory. The added benefit of CNN includes the automatic detection of most exclusionary features among the classes.

Furthermore, future work can be extended in two parts; the first part will cover development of classification system which will classify similar infectious diseases, for instance, SARS, MERS, ARDS, and bacterial Pneumonia using x-ray images, and the second part, will include the training of DL models using generated dataset through the generative adversarial network. More robust models can be developed using a dataset from several sources.

**Table 10. Accuracy obtained by existing models and models used in the study.**

| References | Images Type | No of Images | Method | Accuracy |
|---|---|---|---|---|
| Ozturk et al. [10] | Chest x-ray | 125COVID-19 / 500Normal | DarkCovidNet | 98.08% |
| | Chest x-ray | 125COVID-19/ 500Normal/ 500Pneumonia | DarkCovidNet | 87.02% |
| Narin et al. [8] | Chest x-ray | 50COVID-19 / 50Normal | ResNet50, Deep CNN | 98% |
| Sethey et al. [59] | Chest x-ray | 25COVID-19 / 25Normal | ResNet50 + SVM | 95.38% |
| Ioannis et al. [58] | Chest x-ray | 224COVID-19 / 700Pneumonia / 504Normal | VGG-19 | 93.48% |
| Wang et al. [9] | Chest x-ray | 53COVID-19 / 5526Normal | COVID-Net | 92.4% |
| Hemdan et al. [57] | Chest x-ray | 25COVID-19 / 25Normal | COVIDX-Net | 90% |
| Zheng et al. [62] | Chest CT | 213COVID-19 / 229Normal | UNet+3D Network | 90.8% |
| Ying et al. [60] | Chest CT | 777COVID-19 / 708 Normal | DRE-Net | 86% |
| Xu et al. [18] | Chest CT | 219COVID-19 / 175Normal / 224Pneumonia | ResNet + Location Attention | 86.7% |
| wang et al. [61] | Chest CT | 195COVID-19 / 258Normal | M-Inception | 82.9% |
| **Our Proposed CNN Method** | **Chest x-ray** | 140COVID-19 / 140Normal | **CNN** | **97.62%** |
| | **Chest x-ray** | 140COVID-19 / 140Normal /140 Pneumonia | **CNN** | **93.75%** |
| **Our Employed Pre-trained Method** | **Chest x-ray** | 140COVID-19 / 140Normal | **VGG16** | **100%** |
| | **Chest x-ray** | 140COVID-19 / 140Normal /140 Pneumonia | **VGG16** | **87.50%** |

## Author Contributions

**Conceptualization:** Hua Wang.

**Data curation:** Rubina Sarki.

**Formal analysis:** Rubina Sarki.

**Investigation:** Rubina Sarki.

**Methodology:** Rubina Sarki.

**Resources:** Kate Wang.

**Supervision:** Khandakar Ahmed, Yanchun Zhang.

**Validation:** Kate Wang.

**Writing – review & editing:** Hua Wang, Kate Wang.

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
