## [Decision Letter · Decision Letter 0]

9 Nov 2021

PONE-D-21-24533

Automated Detection of COVID-19 through Convolutional Neural Network using Chest x-ray Images

PLOS ONE

Dear Dr. Wang,

Thank you for submitting your manuscript to PLOS ONE. After careful consideration, we feel that it has merit but does not fully meet PLOS ONE’s publication criteria as it currently stands. Therefore, we invite you to submit a revised version of the manuscript that addresses the points raised during the review process.

We look forward to receiving your revised manuscript.

Kind regards,

Xiaodi Huang, PhD

Academic Editor

PLOS ONE

Journal Requirements:

5. Please note that in order to use the direct billing option the corresponding author must be affiliated with the chosen institute. Please either amend your manuscript to change the affiliation or corresponding author, or email us at plosone@plos.org with a request to remove this option.

Additional Editor Comments:

Please address the reviewers' questions one by one.

The revised manuscript must be  the proof-read.

Reviewers' comments:

Reviewer's Responses to Questions

**Comments to the Author**

1. Is the manuscript technically sound, and do the data support the conclusions?

Reviewer #1: Yes

Reviewer #2: Yes

2. Has the statistical analysis been performed appropriately and rigorously? 

Reviewer #1: Yes

Reviewer #2: Yes

3. Have the authors made all data underlying the findings in their manuscript fully available?

Reviewer #1: Yes

Reviewer #2: Yes

4. Is the manuscript presented in an intelligible fashion and written in standard English?

Reviewer #1: Yes

Reviewer #2: Yes

5. Review Comments to the Author

Reviewer #1: This paper aims to develop a deep learning-based system for the persuasive classification and reliable detection of COVID-19 using chest radiography. Authors evaluate the performance of various state-of-the-art convolutional neural networks (CNNs) proposed over recent years for medical image classification. They have developed and trained CNN from scratch. A recently published public X-Ray dataset for training and validation purposes is used. The comprehensive ROC and confusion metric analysis with 10-fold cross-validation strongly demonstrate the ideas and methods developed in the paper. Experiment results are compared to demonstrate the effectiveness of models in classification scenarios and their potential for COVID-19 classification, detection, prevention, and control.

The paper is well organised with a clear motivation, innovation and also rich experiments. One minor suggestion is to have two paragraphs in the final section for conclusion and future work.

Reviewer #2: In this manuscript, authors have proposed and presented a CNN oriented COVID-19 detection method, which combine transfer learning and CNN-based model and achieves satisfied results. The idea is interesting. The intention of this study is to evaluate Deep Learning methods for the diagnosis of COVID-19 from X-Ray images. Specifically, both state-of-the-art pretrained CNNs (e.g., VGG) and hand-crafted CNN are utilized either for transfer learning or training from scratch. A relatively small image dataset (at least for the COVID-19 class) is utilized and the results confirm the findings of several related research.

Nevertheless, there are some issues I would like to point out to enrich the contribution, and the proposal description, which I think must be attended to:

• Authors must highlight the contribution in a more profound way to identify the analysis of the several sources consulted.

• Please explain what is CNN II “Therefore, we build a new CNN II which yielded accuracy 97.62% in binary classification” in section 4.2. Probably authors meant Scenario II. Clarify it.

• In Table 7, the necessary space is missing, which is a little confusing. If possible, rearrange the Table 7.

• More details need to be presented to show the significance of the results presented in Table 8 and 9.

• Finally, there are some grammatical and structural errors that need to be rectified in the revised version of this manuscript.

Overall, good work has been presented.

6. PLOS authors have the option to publish the peer review history of their article (what does this mean?). If published, this will include your full peer review and any attached files.

Reviewer #1: **Yes: **Jianming Yong

Reviewer #2: No

---

## [Author Response · Author response to Decision Letter 0]

10 Dec 2021

We greatly appreciate your feedback on how to improve the quality of our manuscript ‘Automated Detection of COVID-19 through Convolutional Neural Network using Chest x-ray Images.” Please find below the point-by-point responses to any concerns raised by the reviewers. The required modifications have been made in the article as suggested. Thank you once again for your constructive comments and opportunity to resubmit again. 

Detailed response to reviewers is attached in the submission.

---

## [Editor Report · Decision Letter 1]

16 Dec 2021

Automated Detection of COVID-19 through Convolutional Neural Network using Chest x-ray Images

PONE-D-21-24533R1

Dear Dr. Wang,

We’re pleased to inform you that your manuscript has been judged scientifically suitable for publication and will be formally accepted for publication once it meets all outstanding technical requirements.

Kind regards,

Xiaodi Huang, PhD

Academic Editor

PLOS ONE

Additional Editor Comments (optional):

I have read the modified version of this submission. All the minor issues raised by the reviewers have been addressed.

Therefore, I recommend to publish this paper.
---

## [Editor Report · Acceptance letter]

26 Dec 2021

PONE-D-21-24533R1 

Automated Detection of COVID-19 through Convolutional Neural Network using Chest x-ray Images 

Dear Dr. Wang:

I'm pleased to inform you that your manuscript has been deemed suitable for publication in PLOS ONE. Congratulations! Your manuscript is now with our production department. 

Kind regards, 

on behalf of

Dr. Xiaodi Huang 

Academic Editor

PLOS ONE